# Effects of Population Declines on Habitat Segregation and Activity Patterns of Rabbits and Hares in Doñana National Park, Spain

Juan F. Beltrán [1,*], Jaime R. Rau [2], Ramón C. Soriguer [3], Maura B. Kufner [4], Miguel Delibes [5] and Francisco Carro [3]

[1]  Departamento de Zoología, Universidad de Sevilla, 41012 Seville, Spain
[2]  Laboratorio de Ecología, Departamento de Ciencias Biológicas y Biodiversidad, Universidad Los Lagos, Campus Osorno, Osorno 5290000, Chile; jrau@ulagos.cl
[3]  Departamento de Etología y Conservación de la Biodiversidad, Estación Biológica de Doñana, CSIC, 41092 Seville, Spain; soriguer@ebd.csic.es (R.C.S.); pcarro@ebd.csic.es (F.C.)
[4]  Center Dr. Ricardo Luti of Ecology and Renewable Natural Resources Research (CERNAR), National University of Córdoba, Córdoba X5000GYA, Argentina; beti0611@gmail.com
[5]  Departamento de Biología de la Conservación de la Biodiversidad, Estación Biológica de Doñana, CSIC, 41092 Seville, Spain; mdelibes@ebd.csic.es
*   Correspondence: beltran@us.es

**Abstract:** Competition, predation, and diseases are key factors shaping animal communities. In recent decades, lagomorphs in Europe have been impacted by virus-borne diseases that have caused substantial declines in their populations and, subsequently, in many of their predators. We examined activity and habitat-use patterns of sympatric European rabbits (*Oryctolagus cuniculus* L.) and Iberian hares (*Lepus granatensis* R.) in Doñana National Park, Spain, (DNP) during two periods of disease outbreak. In the first period (1984–1985), fecal pellet counts and roadside counts indicated that lagomorph species were segregated, with rabbits occurring in scrublands and hares in marshlands. Both species also occupied rush and fern belt ecotones. Roadside counts at sunrise, midday, sunset, and midnight revealed that rabbits and hares had the same activity patterns (crepuscular and nocturnal) in the zone of sympatry. During the second period (2005–2016), roadside counts showed that rabbits and hares were mainly nocturnal in scrublands and border marshlands. Hares occupied scrublands; a habitat previously occupied only by rabbits. These results are interpreted in light of the competition theory and predation pressure. The disease-caused decline of rabbits has likely favored hares that moved into scrublands, a vegetation type previously occupied exclusively by rabbits. The decline of rabbits in DNP has also caused the almost disappearance of this area of the Iberian lynx (*Lynx pardinus*), a rabbit specialist, thus enabling generalist predators to increase. Generalist predators have subsequently increased predation pressure on both rabbits and hares, causing them to switch to nocturnal activity.

**Keywords:** activity patterns; *Lepus granatensis*; population decline; niche; *Oryctolagus cuniculus*; roadside census; predator–prey relationships; spatio-temporal behavior

## 1. Introduction

Competition for resources is considered one of the main factors shaping the coexistence of species in natural communities [1]. Among lagomorphs, competition is expected to be greater among closely related species, either phylogenetically or ecologically [2,3]. Predation can also play a role in how species co-exist by increasing mortality or modifying spatio-temporal activity patterns of lagomorphs [4–6]. Additionally, diseases may affect community structure, especially if they impact a keystone species [7–9]. In Europe, populations of rabbits and hares have been affected by imported diseases that have substantially

reduced their populations, and subsequently caused a reduction of predators dependent on them [9,10].

After the emergence of myxomatosis in 1950s, rabbit (*Oryctolagus cuniculus*) populations recovered, only to crash again in late 1980s from rabbit hemorrhagic disease (RHD) [7,11]. In several countries, the first epizootic occurrence of myxomatosis was accompanied by an increase in European hare (*Lepus europaeus*) populations, followed by a concomitant decline of European rabbits [12–18]. This pattern was interpreted as indirect evidence of competition between both species [19–22]. Overall, three possible mechanisms of competition between rabbits and hares have been documented: (a) diseases that harm hares, but not rabbits (e.g., the stomach worm (*Graphidium strigosum*) [17]); (b) despite the known aggressiveness of hares [23], behavioral observations that suggested rabbits are the "winners" when directly fighting hares [21], or rabbits driving hares away from their burrows (but see Broekhuizen [16]); and (c) competition for food [24].

European rabbits were impacted again in 2010 [25,26] with the arrival of a novel genotype of the calicivirus RHDV (RHDV2 or RHDVb or *Lagovirus europaeus*), ref. [27] that reached Spain in 2011 [9], and Doñana National Park (DNP) more specifically in 2013. A long-term monitoring program at DNP detected a decline in rabbit numbers during 2013. In Coto del Rey (northern DNP), there was a decline of >80% of rabbits during 2012–2013 [28]. Similar declines were detected in all populations surveyed within DNP (F. Carro, unp. obs.). On the other hand, the Iberian hare population in the DNP underwent a moderate decline in the period 1996–2012 that was attributed to varying flooding cycles of the marshlands, changes in vegetation cover, and predation pressure [29]. Since 2003, hare numbers have decreased by 88% in DNP [29], and they were also affected by the first outbreaks of the novel ha-MYXV in 2018 [30].

Interspecific competition can be indirectly assessed from both allopatric and contiguous spatial distributions [31–33]. To minimize competition, potential competitors segregate along predictable niche parameters of diet, habitat use, and activity periods [31]. Therefore, the effects of myxomatosis and RHD among sympatric populations of hares and rabbits were expected to not only go beyond changes in their respective abundance patterns, but also to affect habitat use and activity periods of both lagomorphs. Long-term studies may shed light on vertebrate population dynamics [34]. In particular, changes in abundance, especially when those changes affect species differentially, may be seen as a "natural experiment" [35].

In this paper, we use a natural experiment to examine niche relationships between rabbits and hares in southwestern Spain. Our objective was to evaluate the effects of rabbit population crashes (mid 1980s and 2011–2013) on hare populations, specifically if habitat use and activity patterns varied in the context of the competitive exclusion hypothesis. We speculated that the collapse of rabbit populations would not only reduce competitive interactions between rabbits and hares, but also increase predator pressure on remaining lagomorphs. Specifically, we expected hares to expand into habitats previously occupied only by rabbits [3]. We also expected changes in activity, with rabbits becoming nocturnal in response to increased predation pressure [4].

## 2. Materials and Methods

### 2.1. Study Area

Our study was conducted in Doñana Biological Reserve (DBR, Figure 1), a restricted area of the Doñana National Park (DNP), situated on the right bank of the mouth of Guadalquivir River (approximately 37° N, 6°30′ W), spanning ≈ 1220 km$^2$, including the peripheral zone of protection. The marshland or "marisma" is usually flooded from October–November to May–June, and spans ≈ 55% of the DNP area. Around 30% of DNP is Mediterranean scrubland, and 15% of DNP is sand dunes with scattered pine forest (*Pinus pinea*). A detailed description of the DNP area can be found in Valverde (1958) [36], Aguilar-Amat et al. (1979) [37], and Green et al. (2018) [38]. The climate is Mediterranean

with a slight Atlantic influence. Summers are warm and dry, and winters mild and wet. The average annual rainfall is 500–600 mm, 87% of which falls from October to April.

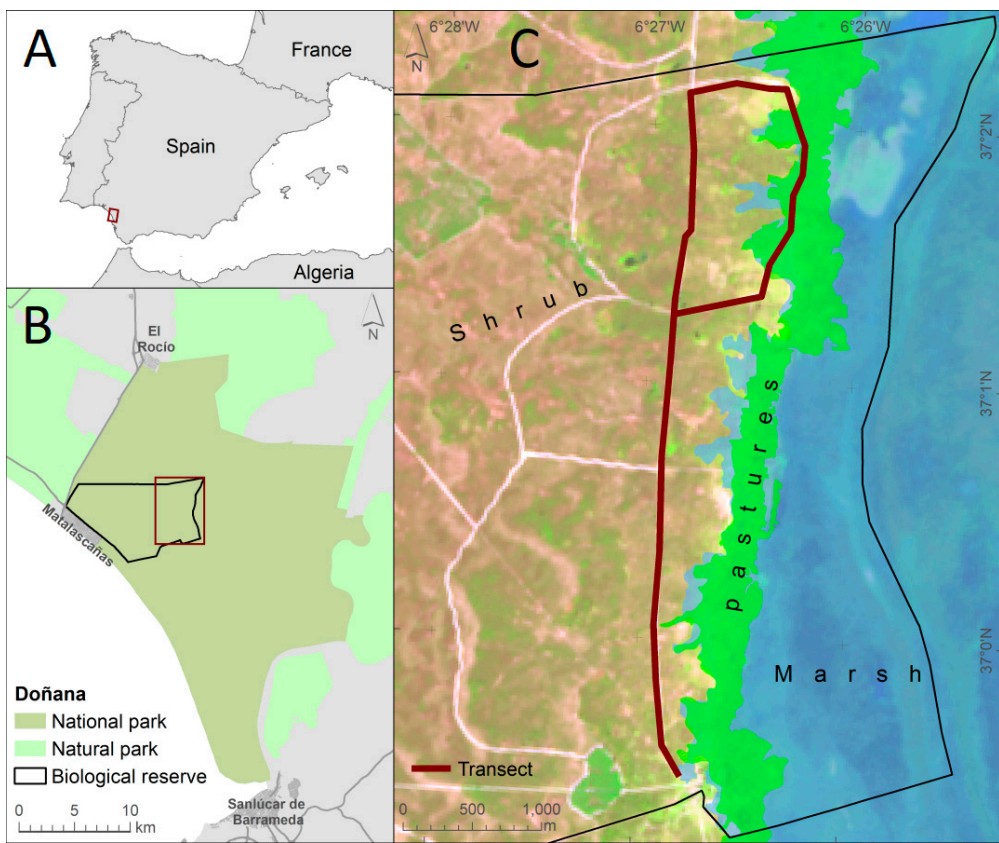

**Figure 1.** Location of the study area (Biological Reserve of Doñana) in Doñana National Park ((**A**,**B**), southwest Spain). The transect ((**C**), ~8 km long) used for roadside counts of lagomorphs.

Vegetation types present include pure scrublands, border scrublands, a fern belt, a rush belt, border marshlands, and pure marshlands (Figure 1) [39,40]. Each one of the four transitional bands in the ecotone (known locally as "La Vera") ranges from nearly 15 m to more than 120 m wide.

### 2.2. First Period of the Study: 1983–1985

The first period of the study was characterized by a pre-epizootic decline of rabbits and hares. Beginning in early November 1983 (just before the flooding of the *marisma* that year), 10 plots of 1 m² spaced 1 m apart were randomly placed in each of the six vegetation bands. To assess spatial distributions of rabbits and hares, all lagomorph fecal pellets were counted and removed from each plot (see [41] for a review). In addition, in August 1984 (mid-summer), 10 similar plots were established in a pure marsh area (Leo Biaggi area) approximately 10 km east of the ecotone (Figure 1). Hare pellets are usually larger than those of rabbits [42,43]. To differentiate between them, we collected samples of fresh pellets in rabbit warrens (January 1984) and hare bedding sites (October 1983), and recorded three measurements (thickness = minimum diameter, width, and length = maximum diameter) using a caliper. To assess the frequency of both species within each sampling plot, we used a filter approach based on the three measurements to differentiate pellets of rabbits from those of hares.

Roadside counts of both lagomorphs [44,45] were conducted along a 5.2 km transect in scrublands and a 2.8 km ecotone transect (hereafter called border marshland transect; Figure 1) using a 4 × 4 vehicle that traveled under 20 km/h. Counts were made at sunset, approximately four times a month from November 1983 to November 1984. The abundance of

hares and rabbits was indexed by individuals tallied/10 km driven (kilometric abundance index, KAI). To examine circadian activity, roadside counts of both species were performed four times a day at sunrise, midday, sunset, and midnight for three consecutive days during the winter of 1985, along the previously described transects. To express lagomorph abundance on a comparative biomass basis, average autumn weights of both species in the study area (both sexes pooled, rabbits, $n$ = 20, mean $\pm$ standard error = 0.91 $\pm$ 0.04 kg; hares, $n$ = 18, m $\pm$ SE = 1.98 $\pm$ 0.06 kg) were obtained from the records of the Doñana Biological Station scientific collections.

Statistical analyses followed the procedures of Zar [46]. We generally considered equal sample sizes with two-tailed hypotheses, therefore the *t*-tests (both paired-sample *t*-test and two-sample *t*-test) were used. However, when equal sample size or other requirements could not be met, nonparametric equivalents were used.

### 2.3. Second Period of Study: 2005–2016

The second period of the study occurred from 2005–2007 and 2014–2016 when populations of both lagomorphs had suffered dramatic declines. Hares and rabbits counts (individuals/10 km, KAI) were obtained from the same transects used in 1983–1985. Surveys were also conducted via a 4 × 4 vehicle, both at sunset (from 1.5 h before sunset to sunset) and at night (1.5 h after sunset) with the aid of a handheld 100-watt spotlight after dusk [47,48]. Surveys were conducted in spring, summer, and early autumn during 2005 and 2007, and in spring and autumn during 2014 and 2016 by at least two people [49–51]. Observers were seated on the roof of the vehicle ≈ 3 m above ground level [52]. Hares and rabbits were identified using binoculars. We also measured the perpendicular distance of animals from the transect line using a laser telemeter. The maximum width of the contact strip was 200 m, and depended on the height of the vegetation at the time. The majority of contacts occurred within 100 m of the observer. Pellet counts were performed only during the first period of study, and were used to examine habitat segregation. Roadside counts were used complementarily, both in the first and the second period, as an efficient method to assess not only habitat segregation, but also estimate population trends [7,44,53].

## 3. Results
### 3.1. First Period: 1983–1985
#### 3.1.1. Pellet Characteristics

Pellet-size distributions resembled normality, enabling us to calculate 95% confidence intervals (Figure 2). Hare pellets were longer (t = 3.9, $p$ < 0.01), wider (t =11.1, $p$ < 0.01), and thicker (t = 12.2, $p$ < 0.01) than rabbit pellets, and our measurements were consistent with those reported elsewhere [42,43]. However, overlap did occur in thickness (24%), therefore we used 7 mm to distinguish rabbit (≤7 mm) from hare (>7 mm) pellets, but the overlap of both species at 5–7.5 mm was evident (Figure 2). Hence, we distinguished "probable rabbit pellets" from "probable hare pellets".

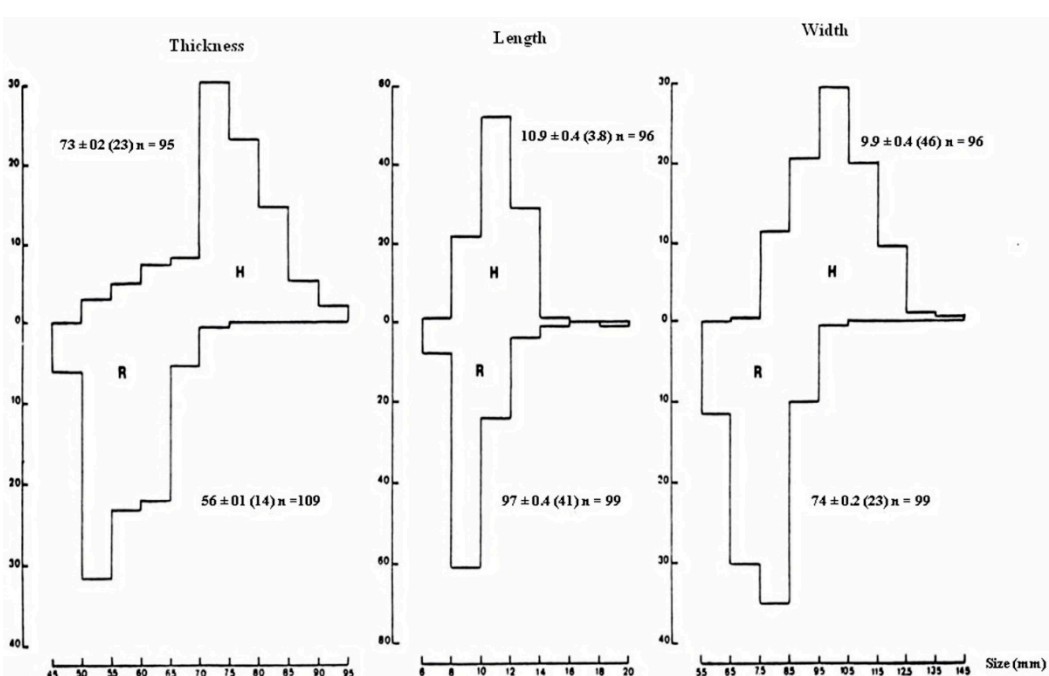

**Figure 2.** Frequency histograms of Iberian hare (above, H) and European rabbit (below, R) pellet dimensions from samples collected in rabbit warrens (January 1984 and hare bedding sites (October 1983). The mean ± SE, 95%, confidence intervals, and sample sizes are indicated.

### 3.1.2. Habitat Use

Based on species-specific traits of fecal pellets (Figure 2), habitat use differed by species (Figure 3). Hares mainly used marshlands, and rabbits scrublands. Both species used transitional zones in the ecotone in the same proportion (i.e., there were no significant differences). This habitat segregation was also confirmed when an independence analysis was applied to the marshland and scrubland pellet abundances ($2 \times 2$ contingency test corrected for continuity; $p < 0.001$). Additional pellet counts made at the marshland habitat (see Section 2.1 and Section 2.2) did not detect rabbit pellets, while $3.0 + 0.98$ hare pellets/m$_2$ were found.

Roadside counts indicated that rabbit abundance was four times higher in the scrubland than in the ecotone (Table 1: the means differ statistically; paired sample *t*-test; t = 3.3, $p = 0.02$). Hares did not occupy scrublands during the first period of study, and both species were observed in the ecotone (Table 1: paired-sample *t*-test; t = 3.3, $p = 0.02$). In this zone, peaks in hare abundance appeared to be related to the start of the marsh flooding (e.g., November 1983), and probably linked to the beginning of the breeding season (April 1984). By considering only the six months in which both species occur together each year in the study area (Table 1), no statistical differences were found between the abundances of rabbits and hares in the ecotone (paired-sample *t*-test; t = 0.41, 220 $p = 0.69$), but when biomass was considered (Table 1), hare biomass was 2.6 times greater than rabbit biomass.

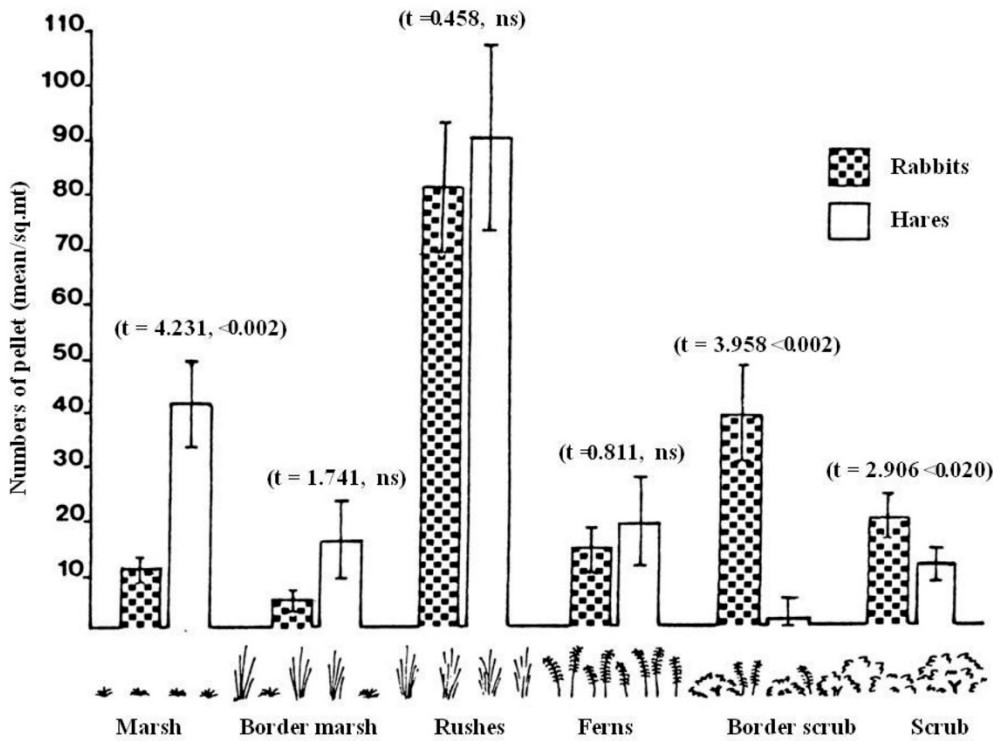

**Figure 3.** Habitat use by European rabbits and Iberian hares in six vegetation zones at Doñana National Park, SW Spain. Statistical *t*-values are indicated for each vegetation type and are provided at the top of the bars. Note that transitional zones were used by both species, whereas hares were more abundant in the marshland and rabbits in the scrubland.

**Table 1.** Relative abundances based on roadside counts (individuals/10 km) of European rabbits and Iberian hares in two vegetation types in Doñana National Park, 1983 and 1984. Means ± SE and sample size are indicated for each month.

| | | Scrublands | Border Marshlands | |
|---|---|---|---|---|
| **Month** | *n* | **Rabbits** | **Rabbits** | **Hares** |
| November 1983 | 3 | 10.7 ± 8.7 | 8.3 ± 3.3 | 45.0 ± 35.5 |
| December 1983 | 2 | 33.0 ± 19.1 | 0.0 | 2.5 ± 2.5 |
| April 1984 | 3 | 32.7 ± 10.5 | 11.7 ± 9.3 | 18.3 ± 8.3 |
| May 1984 | 4 | 66.5 ± 18.0 | 2.5 ± 2.5 | 5 ± 3.5 |
| July 1984 | 4 | 67.0 ± 15.7 | 27.5 ± 7.8 | 1.2 ± 1.2 |
| November 1984 | 3 | 18.7 ± 4.4 | 3.3 ± 3.3 | 1.7 ± 1.7 |
| Grand mean | 19 | 41.4 ± 7.4 | 10.0 ± 3.1 | 11.8 ± 6.1 |
| Average biomass(kg/10 km) | | 37.7 | 9.1 | 23.4 |

### 3.1.3. Circadian Activity

In the border marshland where both species occurred during winter, rabbits and hares were both most active at dusk and night (Figure 4). However, in the scrublands, rabbits appeared to be most active from midday to sunset, at least in winter; few individuals were seen active at night.

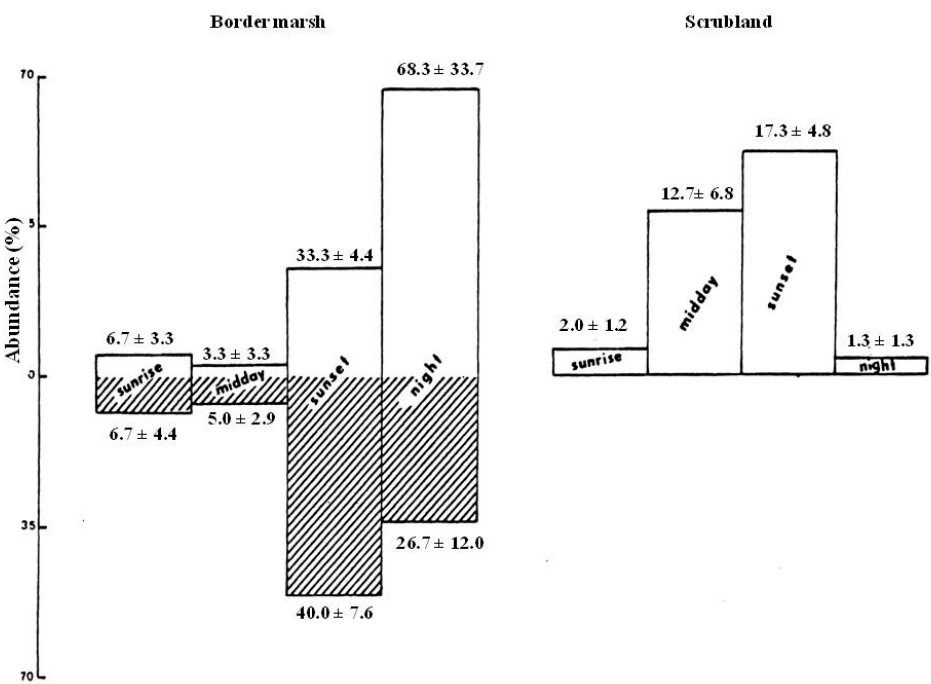

**Figure 4.** Circadian activity of European rabbits (white bars) and Iberian hares (stripped bars) at the border marshland and scrubland zones, estimated based on roadside counts in four time intervals over three consecutive days in winter 1985. Means ± SE of individuals/10 km are provided, and pairwise comparisons by species within a time period were not significant.

*3.2. Second Period: 2005–2016*

After the sharp decline of rabbits, changes in habitat use were detected. In scrublands, the peak number of rabbits was observed at sunset in 1985 (Figure 4), but this changed to night in 2005–2006 (Tables A1 and A2, Figure 5). However, small sample sizes limited these comparisons. In border marshlands, where rabbits showed a mainly nocturnal pattern in 1985, they seemed to maintain this pattern (t = 1.77, *p* = 0.10).

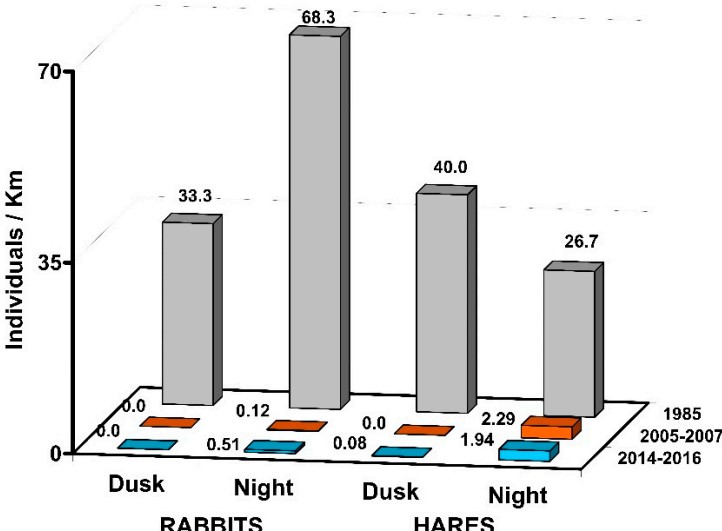

**Figure 5.** Abundances of European rabbits and Iberian hares in the border marshland during sunset and night, as estimated by roadside counts in the three intervals of study (data from 1985 are the same as in Figure 4; the rest of the data are from Tables A1 and A2). Note that, in spite of the two-order of magnitude drop in population numbers since 2005, rabbits have maintained the same pattern of activity (being mostly nocturnal in this habitat).

For hares, observed changes included both habitat use and activity patterns. In the first study period (1984–1985), no hares were observed in the scrublands, and their activity peaked in the border marshlands at sunset (Figure 4). From 2005 to 2006, hares tended toward a nocturnal activity (Figure 5), not only in marshlands (although differences there were not significant, t = 0.14, *p* = 0.69), but also in the scrublands, where they had been absent during the roadside counts in 1984–1985.

## 4. Discussion

In the initial study period, rabbits and hares were spatially segregated. This could indicate that exploitative competition occurs between both species [54]. The distinct associations of rabbits with scrublands and hares with marshlands may be one of the factors enabling the species to coexist [55]. In a previous study, Rogers and Myers [56] also did not detect rabbits in open marshes. Based on resting time (daytime) habitat selection, Vidus-Rosin et al. [56] also reported segregation between European hares (*Lepus europaeus*), and introduced eastern cottontails (*Sylvilagus floridanus*) as a possible strategy to reduce interspecific competition. However, two criticisms can be made regarding the results on differential habitat use based on fecal-pellet counts. First, our data correspond to "probable" rabbit and hare pellets, since no definite distinction of pellet size was found. Second, Rowland et al. [57] suggested that fecal pellet distributions may not be directly associated with habitat use by herbivores due to the patchy distribution of foraging areas.

Using sand-track records in summer, Alvarez et al. [58] reported that the circadian activity of rabbits in Doñana in the scrublands was mainly crepuscular and nocturnal. Similarly, Villafuerte et al. [59], using roadside counts in scrublands, also showed that the majority of rabbit activity occurred during twilight periods and night. High levels of rabbit activity at sunset were constant throughout the year. When occupying the same vegetation type (border marshlands), our observations indicated that rabbits and hares foraged at the same time (sunset-night), thus increasing the probability for interference competition [60]. Within border marshlands, grasses and forbs account for ~70% of the diet of rabbits in Doñana [61]. Since this foraging area is spatially (see Figure 1) and temporally limited (especially when the nutritional value of grasses is taken into account [62]), interspecific competition can result in a species–resource specialization [63], where rabbits became scrubland specialists and hares became marshland specialists.

The observation that rabbits and hares use different but neighboring habitats (i.e., contiguous allopatry) when one species is dominant to the other would be an example of "type 1 coexistence" (*sensu* [27]), and enables long-term coexistence. Several other factors, including disease outbreaks or differential predation, could modify the competitive superiority of rabbits [17,21]. An infected rabbit would be easier for predators to capture than a healthy hare. Predators may then modify patterns of coexistence of rabbits and hares by exerting greater predation pressure on rabbits [64]. Specifically, the coexistence of rabbits and hares may have been mediated by specialist predators upon rabbits, such as the Iberian lynx (*Lynx pardinus* T.) and the Iberian imperial eagle (*Aquila adalberti* B.) [65,66]. Further, the abundance of hares in DNP may have been controlled by such factors as seasonal habitat reduction (i.e., the flooding of the marshland [67]), direct interference by rabbits (i.e., agonistic interactions [21]), and infestation by stomach parasites (17).

The situation and interpretation described above was unexpectedly tested with the decline of the population numbers of both rabbits (due to the arrival of two successive epizootics of RHDV) and hares (mainly due to the varying flooding cycles of the marshlands and increased predation pressure [53]). This "natural experiment" gave us an opportunity to observe responses at temporal and spatial scales, and evaluate previous interpretations.

Our data show that rabbits have become mainly nocturnal, a behavior previously restricted to a more open border scrubland habitat. This was likely a response to increased predation pressure among remaining rabbits [4,68,69]. Nocturnal activity is also currently observed at a much larger spatial scale (i.e., DNP, Carro et al., unp. obs.), and contrasts to

previous studies that showed rabbits as mostly crepuscular (sunset and dawn) with some activity at night (mostly during February–March) [59].

Among hares, the population response observed was two-fold. First, on a spatial basis, recent (2005–2016) roadside counts showed the presence of hares in scrublands. Hares occupied scrublands year-round, rather than as a seasonal response to flooding of the marshlands in autumn (Tables A1 and A2). Second, circadian activity by hares also changed from crepuscular to nocturnal (Figure 4). Such patterns have been reported via the use of radio-collared hares [70]. Contrary to reported by Katona et al. [24], we did not observe an increase in the hare abundance in the study area.

The implications of the dramatic declines of rabbits and hares in the Doñana ecosystem are likely complex. Perhaps most striking are implications to endangered species in the area. The Iberian lynx (*Lynx pardinus*) is a rabbit specialist, with rabbits representing 85–95% of their diet [66]. As a result of the reduced rabbit abundance, the lynx population in DNP has nearly been extirpated. A similar fate is threatening the Iberian imperial eagle (*Aquila adalberti*) [71]. This pattern of severe decline contrasts with generalist carnivores in the area, including red foxes (*Vulpes vulpes*), badgers (*Meles meles*), and mongooses (*Herpestes ichneumon*). These predators are able to switch among available food sources, making them resilient to the decline of rabbits. Field observations suggest that the fox abundance in DNP has increased in recent years (F. Carro, *unp. obs.*). This increase may have been a response to the reduction in control of foxes by lynxes or mesopredators [72].

## 5. Conclusions

We conclude that, after the population decline of both species, their ecological overlap has increased substantially. Their convergence in nocturnal behavior is most likely a response to the increase in predation pressure [73,74]. Most recent numbers (2005–2016) of roadside counted rabbits are two orders of magnitude lower than in 1985, and hare numbers are only slightly higher (Appendix A, Tables A1 and A2). The remaining individuals of both species have selected the night as the circadian period that presents the minimum risk of predation. Bakker et al. [75] have shown that European rabbits are sensitive to perceived predation risk [76]. Most of the terrestrial predators of rabbits and hares in DNP have activity peaks at sunset and dawn [4,68,77].

The observed move by hares into vegetation types exclusively inhabited by rabbits supports the implications of relaxing competitive exclusion. Research focusing on both species at sympatry (e.g., the Doñana ecotone), allopatry (rabbits and hares feeding only in their respective preferred habitats), and removal experiments [31] will be valuable. Our study shows the importance of maintaining long-term monitoring of wildlife populations using standard procedures. This endeavor is enormously facilitated in protected areas, and needs the commitment and effort of their people.

**Author Contributions:** Conceptualization, methodology, and field sampling, J.R.R., J.F.B., F.C., R.C.S., M.D. and M.B.K.; resources, J.R.R., J.F.B., F.C., R.C.S., M.B.K. and M.D.; writing—original draft preparation, J.R.R. and J.F.B.; writing—review and editing, J.F.B., J.R.R., F.C., M.D. and R.C.S. All authors have read and agreed to the published version of the manuscript.

**Funding:** During the study, J.F.B. enjoyed a predoctoral fellowship from the Consejo Superior de Investigaciones Científicas (CSIC), and J.R.R. and M.B.K enjoyed a grant from the Instituto de Cooperación Iberoamericana. Financial assistance for the trip of M.D. to Canada was received from the Consejería de Educación de la Junta de Andalucía. The first period of study (1983–1985) was included in the project 944 of CSIC-CAICYT. During the second study interval (2005–2016), this research was partially supported by the Dirección General de Espacios Naturales y Participación Ciudadana, Consejería de Medio Ambiente y Ordenación del Territorio, Junta de Andalucía. The authors would like to thank the EBD-CSIC monitoring team and Doñana ICTS-RBD, who provided logistic and technical support.

**Institutional Review Board Statement:** Not applicable.

**Informed Consent Statement:** Not applicable.

**Data Availability Statement:** Not applicable.

**Acknowledgments:** J.R.R. acknowledges J.E.C. Flux and D. Fraguglione for sending their stimulating papers on lagomorph biology, and to his wife, Angélica Catalán, for her continuous assistance while producing this research, and also for her patient work with some of the figures of the paper. J. Castroviejo gave us valuable information about Iberian hares' movements when the marshland was flooding. A reduced version of the results of the first period of study (1983–1985) was presented as a poster in the Symposium on Lagomorph Population Dynamics and Relevance to Management (IV International Theriological Congress; 1985, Edmonton, Canada). We are in debt to J.A. Litvaitis, who kindly revised earlier versions of the manuscripts, and made suggestions to improve its content and readability.

**Conflicts of Interest:** The authors declare no conflict of interest.

## Appendix A

Tables A1 and A2. Relative abundances (individuals/10 km) of European rabbits and Iberian hares at two vegetation zones (A1: Ecotone scrubland- marshland, A2: Scrubland)of Doñana National Park, as obtained by roadside counts at sunset and night on the same transect as in Table 1, in two recent intervals of continuous population monitoring. For each month, the mean $\pm$ SE is given, *n* = sample size.

**Table A1.** Ecotone scrubland-marshland.

| Year | Month | Rabbits | | Hares | |
|---|---|---|---|---|---|
| | | **Sunset** | **Night** | **Sunset** | **Night** |
| 2005 | April | 0.00 | 0.70 | 0.35 | 0.35 |
| | June | 0.70 | 0.70 | 0.00 | 1.76 |
| | September | 0.00 | 0.00 | 0.00 | 0.00 |
| **Mean $\pm$ SE** | | **0.23** | **0.47 $\pm$ 0.02** | **0.12** | **0.70 $\pm$ 0.54** |
| 2006 | March | 0.35 | 0.00 | 0.00 | 0.35 |
| | June | 0.00 | 1.60 | 0.00 | 1.40 |
| | September | 0.35 | 0.00 | 0.00 | 0.00 |
| **Mean $\pm$ SE** | | **0.23 $\pm$ 0.12** | **0.35** | **0.00** | **0.58 $\pm$ 0.42** |
| 2007 | March | 0.35 | 1.06 | 0.00 | 6.33 |
| | June | 0.35 | 0.00 | 0.32 | 3.52 |
| | September | 0.00 | 1.06 | 0.00 | 2.81 |
| **Mean $\pm$ SE** | | **0.23 $\pm$ 0.12** | **0.71 $\pm$ 0.35** | **0.12** | **4.22 $\pm$ 1.10** |
| **3-year mean $\pm$ SE** | | **0.23** | **0.51 $\pm$ 0.11** | **0.08 $\pm$ 0.04** | **1.84 $\pm$ 1.19** |
| 2014 | March | 0.00 | 0.00 | 0.00 | 2.11 |
| | September | 0.00 | 0.35 | 0.00 | 0.35 |
| **Mean $\pm$ SE** | | **0.00** | **0.18** | **0.00** | **1.23 $\pm$ 0.88** |
| 2015 | March | 0.00 | 0.00 | 0.00 | 5.27 |
| | September | 0.00 | 0.35 | 0.00 | 0.00 |
| **Mean $\pm$ SE** | | **0.00** | **0.18** | **0.00** | **2.64** |
| 2016 | March | 0.00 | 0.00 | 0.00 | 3.87 |
| | September | 0.00 | 0.00 | 0.00 | 2.11 |
| **Mean $\pm$ SE** | | **0.00** | **0.00** | **0.00** | **2.99 $\pm$ 0.88** |
| **3-year mean $\pm$ SE** | | **0.00** | **0.00** | **0.00** | **2.29 $\pm$ 0.54** |

**Table A2.** Scrubland.

| Year | Month | Rabbits | | Hares | |
|---|---|---|---|---|---|
| | | Sunset | Night | Sunset | Night |
| 2005 | April | 0.00 | 0.57 | 0.00 | 0.00 |
| | June | 0.19 | 0.19 | 0.00 | 0.00 |
| | September | 0.00 | 0.00 | 0.00 | 0.00 |
| **Mean ± SE** | | **0.06** | **0.25 ± 0.17** | **0.00** | **0.00** |
| 2006 | March | 0.00 | 0.00 | 0.00 | 0.00 |
| | June | 0.38 | 0.00 | 0.00 | 0.38 |
| | September | 0.38 | 0.00 | 0.00 | 0.00 |
| **Mean ± SE** | | **0.25 ± 0.13** | **0.00** | **0.00** | **0.13** |
| 2007 | March | 0.00 | 0.19 | 0.00 | 0.19 |
| | June | 0.19 | 0.00 | 0.00 | 0.57 |
| | September | 0.00 | 0.38 | 0.00 | 0.76 |
| **Mean ± SE** | | **0.06** | **0.19 ± 0.11** | **0.00** | **0.51 ± 0.17** |
| **3-year mean ± SE** | | **0.13 ± 0.06** | **0.15 ± 0.08** | **0.00** | **0.21 ± 0.15** |
| 2014 | March | 0.00 | 0.00 | 0.00 | 0.00 |
| | September | 0.00 | 0.77 | 0.00 | 0.77 |
| **Mean ± SE** | | **0.00** | **0.39** | **0.00** | **0.39** |
| 2015 | March | 0.00 | 0.19 | 0.00 | 0.19 |
| | September | 0.19 | 0.19 | 0.00 | 0.00 |
| **Mean ± SE** | | **0.10** | **0.19** | **0.00** | **0.10** |
| 2016 | March | 0.00 | 0.00 | 0.00 | 0.19 |
| | September | 0.00 | 0.19 | 0.00 | 0.19 |
| **Mean ± SE** | | **0.00** | **0.10** | **0.00** | **0.19** |
| **3-year mean ± SE** | | **0.03** | **0.22 ± 0.09** | **0.00** | **0.22 ± 0.09** |

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
