# Peer review of "Effects of Population Declines on Habitat Segregation and Activity Patterns of Rabbits and Hares in Doñana National Park, Spain"

_land, doi:10.3390/land11040461_

Round 1

Reviewer 1 Report

Comments to the Author

The manuscript by Beltrán et al. (Manuscript ID: land-1542145-v1) describes the effects of wild rabbit and Iberian hare population declines on habitat segregation and activity patterns among rabbits and hares in Doñana National Park (DNP, Spain).  The work, carried out in the DNP, is presented by the authors as a long-term study that helped to explore the spatial e temporal niche relationships between European rabbits and Iberian hares in a marshland-scrubland ecotone. Two periods of study where considerer in what seems to be an opportunistic study, namely a first period of study from 1983 to 1985 (3 years), and a second period of study from 2005 to 2016 (11 years). Overall, the work seems interesting and the authors have extensive experience in the area, having published several articles in international journals. However, in this reviewer’s opinion, the present work needs to be improved and reformulated in some aspects in order to be presented more clearly to the reader.

The Materials & Methods section, and the Results along with the discussion section must be reformulated. Taking a better look at the results shown, namely at Table A1, which refers to the “second” period of study, it can be verified that, in fact, the authors are referring to three study periods, of three years each. Hence, the work refers more precisely to 3 trienniums, namely 1983-84, 2005-2007 and 2014-2016. Considering this, there should be a clear division between these three study periods, both in the materials and methods section and in the results section, and the results obtained should be discussed accordingly. Although this work seems to refer to an opportunistic study, in which the experimental design is not always possible to execute perfectly, the authors should, as much as possible, use data obtained in each of the three study periods using the same or similar methodologies so that so that a comparison can be made over time. A few other aspects that need to be revised are pointed out below. Furthermore, the manuscript must be revised by a native English speaker in order to make the speech more fluid and grammatically correct (a few examples are also provided below).

This is why this reviewer cannot recommend the manuscript for publication.

Major revisions:

  1. Lines 17, Abstract section, please replace “..European lagomorphs…” by “…wild lagomorphs from Europe…”
  2. Lines 19, Abstract section, please replace “..sympatric lagomorphs…” by “…sympatric wild lagomorphs…”
  3. Line 21, Abstract section, please replace “…in Doñana National Park, Spain, during the population crash of rabbits (late 1980) and a marked decline of hare numbers (mid 2000).” By “….Doñana National Park, Spain, during two (three?) periods of study, the population crash of rabbits in late 1980 and the marked decline of hare numbers in mid-2000.”
  4. Line 22, Abstract section, please replace “…habitat use of European rabbits and Iberian hares…” by “….habitat use by European rabbits and Iberian hares…”.
  5. Line 24, Abstract section, please replace “…with rabbits in the scrubland and hares in the marsh.” By “….with rabbits occurring in the scrubland and hares in the marsh.”
  6. Line 28, Abstract section, please rewrite the sentence, beginning with “During the second (third and second?) study period(s) (2005-2016), roadside counts….”
  7. Lines 32-33, Abstract section, please rewrite the sentence to: “The decline of wild populations of rabbits and hares have important implications on the composition and relative abundance of predators community.”
  8. Lines 40-41, Introduction section, please rewrite the sentence to: “Among lagomorphs, this competition is expected to be higher between closely related species, either phylogenetically or ecologically.”
  9. Lines 41-43, Introduction section, do the authors mean: “Predation can also play a role in wild species co-existence by increasing the pressure or modulating the spatio-temporal patterns of preys, such as lagomorphs.”?
  10. Line 44, Introduction section, please replace “…especially if they affect keystone species.” By “…especially if impacting on keystone species.”
  11. Lines 45-47, Introduction section, please, rewrite as follows: “In Europe, wild lagomorph populations… and consequently on those of many predator species depending on them.”
  12. Line 47, Introduction section, please replace “…myxomatosis break in 1950s, rabbit…” by “...the emergence of myxomatosis in 1950s, wild rabbit…”
  13. Line 48, Introduction section, please remove “the” from “…in the late…”
  14. Line 49, Introduction section, please write …” rabbit hemorrhagic disease (RHD).”
  15. Line 50, Introduction section, please rewrite “…an increase in the number of European hares (Lepus europaeus) …”
  16. Line 51, Introduction section, please remove the parentheses in the end of the sentence.
  17. Line 52, Introduction section, please replace “…as indirect evidence for competition…” by “…as indirect evidence of competition…”
  18. Line 53, Introduction section, please replace “…mechanisms for competition” by “…mechanisms of competition…”
  19. Line 53, Introduction section, please replace “…both European lagomorphs…” by “…The European wild rabbit and the European brown hare…”, to better specify, once there are other lagomorphs species in Europe.
  20. Lines 53-58, Introduction section, can the authors explain more clearly the competition mechanism indicated in this paragraph? Namely shared diseases that could harm hares, referring to competition via host-parasite relationships, and behavioural aspects since interspecific aggression is documented where rabbits tend to drive hares away from the neighbourhood of their burrows.
  21. Lines 60-62, Introduction section, please rewrite the sentence to: “ Wild rabbit populations from Europe were impacted again in 2010a with the arrival of a new RHD virus, designated RHDV2, RHDVb or Lagovirus europaeus2b, that rapidly spread throughout the Iberian Peninsula reaching Doñana National Park (DNP) in 2013”. a(please see Le Gall-Reculé et al., 2010 and 2013); b(*according to a more recent nomenclature please see Le Pendu et al., 2017)
  22. Line 62, Introduction section, please place the sentence “Iberian hare populations were also heavily affected by the first outbreaks of the novel ha-MYXV in 2018” in the end of the paragraph, in order not to break the information that is provided regarding the wild rabbit.
  23. Line 71, Introduction section, please replace “…the effect of both diseases in sympatric populations of…” by “…the effect of myxomatosis and RHD in sympatric populations of…”
  24. Lines 75-80, Introduction section, please rewrite this paragraph for greater clarity.
  25. Line 81, Introduction section, please replace “…this approach…” by “…the approach explained above…”.
  26. Lines 90-101, Introduction section, this paragraph referring to the authors expectations before the study should be better placed in the discussion section, where the authors could compare the results from the study with their initial expectations, and removed from the introduction section.
  27. Line 105, Materials and Methods section, please replace “…and span ≈ 700 km2 (including the…” by “… spanning ≈ 700 km2, including the…”.
  28. Line 116, Materials and Methods section, please replace “…bands are from about 15 m…” by “…bands range from nearly 15m…”
  29. Line 123, Figure 1. It would be possible to mark on the map the areas related to the four ecotone zones, for a clearer perception of the area occupied by each band?
  30. Line 127-128, Materials and Methods section, please replace “….10 1m2 plots spaced 1 meter apart were stablished randomly in each of the six vegetation bands, …” by “…10 plots with 1m2 spaced 1 meter apart were stablished randomly in each of the six vegetation bands.”
  31. Lines 129-130, Materials and Methods section, please replace “…, and all the droppings of lagomorphs within plots were counted and removed to assess the spatial distribution of rabbits and hares by pellet counts.” By “Whithin plots, all the droppings of lagomorphs were counted and removed to assess the spatial distribution of rabbits and hares by pellet counts.”
  32. Line 137, Materials and Methods section, please replace “…discriminate those of rabbits from hares…” by “…discriminate pellets of rabbits from those of hares…”.
  33. Line 147, Materials and Methods section, did the authors carried out roadside counts of both species four times a day during three consecutive days (total) during the winter of 1985? Please, replace “…using the same transects as said before.” By “…using the transects previously identified.”
  34. Line 152, Materials and Methods section, please provide the meaning of “EBD”.
  35. Line 155, Materials and Methods section, please replace “…t-test (both paired –sample t-test and two-sample t-test) were preferably…” by “…t-test (either paired –sample t-test or two-sample t-test) was preferably…”.
  36. Line 160, Materials and Methods section, please replace “…Iberian hare and rabbits kilometric abundance index…were...” by “…Iberian hare and rabbits kilometric abundance indexes…were...”.
  37. Line 162, Materials and Methods, please replace “…were performed from 1983-85. In our stud...” By “…were performed between 1983-85. In our study…”.
  38. Lines 166-167, Materials and Methods, please replace “…early autumn from 2005 to 2007; in Spring and Autumn from 2014 to 2016, by…” by “…early autumn between 2005 and 2007, and in Spring and Autumn between 2014 and 2016, by…”.
  39. Line 170, Materials and Methods, please replace “We recorded hares and rabbits by direct detection...” by “Hares and rabbits were identified by direct detection…”
  40. Line 174, Materials and Methods, please replace “The majority of contacts were...” by “The majority of contacts occurred…
  41. Line 185, Results section, please replace “…The Iberian hare pellets are longer…” by “…The Iberian hare pellets were shown/ were confirmed to be longer…”
  42. Lines 229-234, Results section, point 3.1.3. Circadian activity, these paragraphs can be merged into a in a simpler and more straightforward paragraph, since differences were found in the mean numbers of both rabbits and hares and both species present a crepuscular-nocturnal behaviour in this area.
  43. Line 240-241, Results section, point 3.1.3. Circadian activity, please move this sentence (“We conclude…crepuscular and nocturnal.”) to the discussion section, framing it in the period of study to which it refers (the first triennium of 1983-85).
  44. Line 245, Results section, please contextualize better the changes in habitat use “After the sharp decline of the European rabbits,…” in DNP, When (study period)? Can the authors identify, for each period of study, the main factors underlying the wild leporids decline (mixomatosis, RHDV, RHDV2…)?
  45. Did the authors carried out pellet characteristics evaluation between 2005-2007 and 2014-2016? If so, these results are not shown.

Reviewer 2 Report

The paper is properly prepared and concerns an interesting problem of lagomorphs in part of Spanish National Park. 

Although, I have some suggestions of changes that I descibed below: 

  1. In keywords I would rather write activity patterns than activity     rhythms (line 35).
  2. In line 32 more proper will be: Besides, rabbits and hares populations' decline have important...
  3. I would change  "spatiotemporal" from line 80 to spatio-temporal like in other fragments.
  4. Line 41 - without "o"
  5. In line 49 - after [7,9] delete double dot.
  6. I would rather write in the past form, like:  we used (line 81), our objectives were (line 83), we expected (line 90), etc.
  7. In line 86 put a space between late 1980, the same like in line 100 between (2003)[24]. 
  8. Signature of figure number 1: please, add to the description A, B, C, because there are three figures in truth. In a legend of figure B, there is a lack of the red line. It should be included. 
  9. I'm not sure what does it 10 1 m2 means in 127 line. Is it correct?
  10. Compare line 142 and 163 and decide which way of descirption is correct: 4x4, or 4 x 4. 
  11. In line 162 it should be study, not stud.
  12. In line 167: seasons of the year we write with small letters, e.g. spring.
  13. In line 185 - were.
  14. Don't put 'son' in empty line. Use text compaction functions in Microsoft Office. The same line 267/268.
  15. Desciption of Figure no 5. Statements starting from: "Note, that..." I would suggest in proper fragment inside the text, not here.
  16. Line 284: lack of ) by (2012.
  17. Line 291: lack of dot.
  18. Line 298: put a space between statements.
  19. Line 301: where is ending bracket [ ) ] ? The same in line 323 and 324 - lack of 2 x ).
  20. Line 364 - red foxes, not Red foxes.
  21. Some remarks to chapter called "References"Lines 436, 489 - lack of dot at the end of lines.Lines 460, 469, 471, 489, 507, 535, 544, 553, - delete pp or Pp.Line 535 - 1981Lines 514 - put short -, not  a longer one – .Line 544: it should be: 1-128, and in 553: 1-11, the same like in 561 line.Lines 563, 539, 518, 443 - Use text compaction functions in Microsoft Office.

Round 2

Reviewer 1 Report

Comments to the Author

In this revised version of the manuscript (Manuscript ID: land-1542145-v2) by Beltrán et al., the authors addressed most of the minor comments and suggestions made in the previous review but, from this reviewer's perspective, the most important aspects remain to be addressed. As previously said, the study carried out by the authors is interesting, but the way it is presented to the reader must be streamlined, especially in the M&M and Results sections.

Major revisions

  1. The M&M section must be streamlined for the sake of clarity and precision. The authors must present the area of study in a subsection, characterize the periods of study in another subsection (time period, pre- or post- epizootic, important events in terms of lagomorph’s populations, indicate that the second period of study integrates two subperiods), and present all the methodologies used in another subsection. This will streamline the M&M section.
  2. For each method uses, authors must indicate, the purpose it serves and here, they can indicate if the method was used in one or in both periods of study. This will avoid the redundancy of describing all the methodologies used in each study period
  3. A correlation between each point of the M&M section and the Results section (e.g. “habitat use”) must also be clearly made, to create a thread for the for a less contextualized reader. Furthermore, if the authors present the results from de first period of study in different sections, the same must be done for the second period of study, so that the reader clearly understands what data are being compared over time.
  4. Data obtained in these two trienniums support the same inferences, as they clarify. In accordance, these explanations should be included in the manuscript, for the sake of clarity and precision. The authors themselves explain these issues assertively in “Authors response”.
  5. Furthermore, it is imperative that the manuscript be revised by a native English speaker, in order to make the speech more fluid and grammatically correct. Despite being requested in the first revision, this aspect was not addressed by the authors.

This is why this reviewer cannot recommend the manuscript for publication.

Minor revisions:

  1. Line 62-64, Introduction section, either the authors consider the novel taxonomy of lagoviruses and consider RHDV2 a novel genotype (Lagovirus europaeus2) or, if opting for the old terminology, must refer to RHDV2 as a novel virus, not as a variant.
  2. Line 62-63, Introduction section, please move …”which reached Spain in 2011…” to line 64 after “…Lagovirus europaeus)…”
  3. Line 64, Introduction section, please replace “Lagovirus europaus” by “Lagovirus europaeus” and cite Le Pendu et al., 2017.
  4. Line 71, Introduction section, the authors must include in the sentence information about the fact that the Iberian hare was not affected neither by classical myxoma virus neither by RHDV but, as they mention, was heavily affected by the novel ha-MYXV. The causes for the decline of the Iberian hare numbers in mid-2000 must be presented better here than in the discussion, where they are provided between brackets.
  5. Lines 79-85, Introduction section, although the authors have tried to improve it, in this reviewer’s opinion, this paragraph it still not clear enough. Please rewrite this paragraph for greater clarity.
  6. Lines 149, M&M section, as they did for the second period of study, the authors should mention that they determined the kilometric abundances Indexes (KAI) for rabbits and hares, so that the less contextualized reader clearly understands that the same indices will be compared in different time periods.
  7. Can the authors submit as supplementary material the table with the statistic relative to figure 2? It would support better the histograms presented.
